

# Multimode-polariton superradiance via Floquet engineering

Christian H. Johansen[1], Johannes Lang[1], Andrea Morales[2],
Alexander Baumgärtner[2], Tobias Donner[2] and Francesco Piazza[1]

**1** Max-Planck-Institut für Physik Komplexer Systeme, 01187 Dresden, Germany
**2** Institute for Quantum Electronics, ETH Zurich, 8093 Zurich, Switzerland

## Abstract

We consider an ensemble of ultracold bosonic atoms within a near-planar cavity, driven by a far detuned laser whose phase is modulated at a frequency comparable to the transverse cavity mode spacing. We show that a strong, dispersive atom-photon coupling can be reached for many transverse cavity modes at once. The resulting Floquet polaritons involve a superposition of a set of cavity modes with a density excitation of the atomic cloud. The mutual interactions between these modes lead to distinct avoided crossings between the polaritons. Increasing the laser drive intensity, a low-lying multimode Floquet polariton softens and eventually becomes undamped, corresponding to the transition to a superradiant, self-organized phase. We demonstrate the stability of the stationary state for a broad range of parameters.



# 1   Introduction

The implementation of strong interactions between photons mediated by a medium [1] is essential for quantum-information processing [2, 3], and at the same time allows for the exploration of the quantum many-body physics of light [4, 5]. Photons inherit interactions from the material through the formation of polaritons [6], hybrid quasiparticles corresponding to an excitation present both in the electromagnetic field and in the material.

The creation of interacting polaritons having access to a macroscopic number of modes is essential for the study of thermodynamic phases of photons and complex types of order [4, 5, 7–15]. The strong-coupling between matter and light, required to implement photon interactions, can be realized by reducing the electromagnetic-mode volume using optical cavities or evanescent fields [1]. The former can also be combined with the use of Rydberg atoms to further enhance interactions [16]. This task obviously becomes more challenging if it needs to be achieved for a whole set of electromagnetic modes, which are in general separated in frequency. One solution is to shape the cavity geometry such that a set of quasidegenerate modes is formed [4, 17–19]. Another option is to use a set of propagating modes as in cavity arrays [14] or photonic crystals [20].

An alternative route towards interacting multimode polaritons has been recently demonstrated experimentally via Floquet engineering of Rydberg levels in a non-degenerate optical cavity [21]. Starting from a given atomic transition resonantly coupled to a single cavity mode, the periodic amplitude modulation of the laser dressing effectively splits the transition into multiple levels separated by the modulation frequency, one of them becoming resonant with a second cavity mode. The resulting two-mode polaritons strongly interact via the Rydberg component inherited from the atomic levels. This demonstrated the potential to Floquet engineer interacting multimode polaritons for the exploration of the many-body physics of light.

In this work, adopting similar ideas, we theoretically study a first example of a phase transition to an ordered phase emerging for multimode Floquet polaritons. Differently from the setup used in [21], the realization of polaritons and their interactions is achieved here via the dispersive coupling between the motional degrees of freedom of an ultracold Bose gas and the transverse electromagnetic (TEM) modes of a near-planar Fabry-Perot resonator [22, 23], with mode-spacing in the GHz range. Similarly to the case of plasmon-polaritons in electronic matter [24], here the polaritons mix a cavity photon with a density excitation in the gas of neutral atoms.

This type of atom-cavity interaction leads to different linear and non-linear susceptibilities compared to [21]. In their case the non-linear susceptibility is dominated by the strongly repulsive Rydberg interaction, which has no counterpart in our system. Coupling to the motional state of the atoms, on the other hand, one can study real-space crystallisation and varying driving schemes makes it possible to program the couplings to the modes supported by the cavity.

If performed at a frequency close to the TEM mode spacing, a periodic phase modulation

of the laser can bring a large number of modes into resonance. This is achieved without heating the atoms, since ultracold density excitations are in the kHz range, and the internal dynamics typically evolves at hundreds of GHz for a correspondingly detuned laser. The mutual interactions between multimode Floquet polaritons induce avoided crossings, controlled not only by the effective light-matter coupling and detunings, but also by the phase-modulation parameters. As a consequence, a low-lying multimode Floquet polariton can be red shifted to zero frequency and subsequently become undamped, corresponding to an instability towards a multimode superradiant phase with macroscopic occupation of the polariton.

The observation of the avoided crossings between coupled Floquet polaritons, in our case, requires sub-recoil resolution achievable in long cavities [23]. Using ultracold bosons coupled to such cavities, single-mode superradiance has already been observed [25], and Floquet modulation has also been studied, but at frequencies far below the transverse mode spacing [26, 27].

Multimode superradiance has been recently observed using ultracold bosons coupled to degenerate confocal resonators [28–31], and is expected to give access to beyond–mean-field effects due to the increased locality of the light-matter coupling [32–36], as well as both static [32] and dynamic [33] frustration. Multimode superradiance is also technologically interesting for implementing quantum models for associative memory, as studied both for confocal cavities [37] and non-degenerate cavities with multi-frequency drive [38].

Floquet superradiance has so far only been studied in the single-cavity-mode case. Here we show that one can enter the multimode regime even in non-degenerate cavities. The number of modes one can actually bring into play with the present Floquet protocol is limited by the achievable amplitudes of phase modulation, which in turn decreases with the modulation frequency. With the types of non-degenerate cavities currently employed in ultracold-atom experiments [39], while interesting multimode nonlinear physics can be realized, it does not seem feasible for our single EOM protocol to reduce the cavity-mediated interaction range significantly below the cavity waist. For this purpose, an extension with multi-frequency drives like [40], which is discussed in section 5.2, should be better suited.

## 2 Theoretical description

### 2.1 Model

We consider a cloud of bosonic atoms trapped inside a near-planar optical cavity and transversally pumped by a laser, as depicted in Fig. 1. This laser is phase modulated (PM) with a frequency comparable to the energy difference between the TEM modes of the cavity. The pump laser is described as a classical field

$$E(t, \mathbf{x}) = 2\lambda \, \eta_p(\mathbf{x}) \cos(\omega_c t + f(t)), \tag{1}$$

with $\omega_c$ being the carrier frequency of the pump, $\lambda$ the pump power and $\eta_p$ its spatial profile. The PM function, $f(t)$, is assumed to be periodic $f(t + T) = f(t)$ and real. Because of the periodicity one can represent $\mathrm{e}^{-if(t)}$ as a discrete Fourier series

$$\mathrm{e}^{-if(t)} = \sum_{\alpha=-\infty}^{\infty} c_\alpha \mathrm{e}^{-i\alpha\Omega t}, \tag{2}$$

where $\Omega = \frac{2\pi}{T}$. In experiments, the PM can be generated using an electro-optical modulator which has a limited bandwidth. In this case the coefficients $c_\alpha$ are functions of the modulation depth $B_m$. With this realization in mind, we consider a series with finite support and define

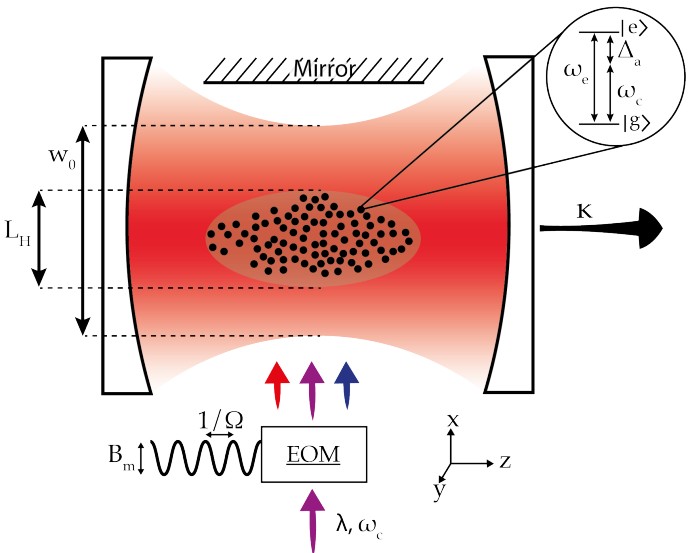

Figure 1: The physical setup considered, consists of a linear cavity with a waist of $w_0$. Inside the cavity ultracold bosonic atoms are confined within a cigar-shaped harmonic trap, with a transverse harmonic oscillator length of $L_H$. Each atom is modelled as a two-level system with excitation energy $\omega_e$. The atoms are transversally pumped with a laser of frequency $\omega_c$ and pump strength $\lambda$. The laser is sent through an electro-optical modulator which introduces a periodic phase-modulation with period $1/\Omega$ and depth $B_m$. Photons are lost from the cavity mirrors with rate $\kappa$, making the system driven and dissipative.

$\alpha_M$ as the index where the sum can be truncated, $c_{|\alpha|>\alpha_M} \sim 0$. The carrier frequency of the pump is red detuned from the atomic transition, $\omega_e$, in such a way that the detuning between atoms and carrier frequency, $\Delta_a$, satisfies $\Delta_a = \omega_e - \omega_c \ll \omega_e + \omega_c$. For instance, for $^{87}$Rb atoms, the $S_{1/2} \rightarrow P_{1/2}$ transition has a frequency $\omega_e \approx 378$ THz [41]. In that case $\Delta_a \sim 10^2$ GHz will easily satisfy the inequality. Similarly, the PM frequency is chosen to be much smaller than $\Delta_a$, such that $\Delta_a + \alpha_M \Omega \sim \Delta_a$. Under these conditions it is well justified to apply the rotating-wave approximation [42].

Since $\Delta_a$ is large compared to the inverse lifetime of the excited state, the occupation of the latter remains small and saturation effects are negligible. Therefore, the ground and excited states of the atoms can be represented as two independent bosonic fields. The resulting Hamiltonian, in the frame rotating with the carrier frequency, in units where $\hbar = 1$, reads

$$
\begin{aligned}
H = \int \mathrm{d}\mathbf{r} \Bigg\{ &\psi^\dagger \left( -\frac{\nabla^2}{2m} + V_g(\mathbf{r}) \right) \psi + \phi^\dagger \Delta_a \phi \\
&+ \phi^\dagger \left( \sum_j g_j \eta_j(\mathbf{r}) a_j + \lambda \eta_p(\mathbf{r}) \mathrm{e}^{-if(t)} \right) \psi + \text{H.c.} + \sum_j \Delta_j a_j^\dagger a_i \Bigg\} .
\end{aligned}
\tag{3}
$$

Here $\psi$ ($\phi$) is the bosonic annihilation field operator for the ground (excited)-state of the atoms with mass $m$. For clarity the space-time dependence of field operators has been suppressed. As $\Delta_a$ is much larger than both kinetic and trapping energy of the excited state, these terms have been neglected. $\eta_j(\mathbf{r})$ is the spatial mode function for the $j$'th cavity mode that couples with strength $g_j$ to the atoms. As the spatial mode functions have explicitly been written in the Hamiltonian, $g_j$ is independent of the mode volume. The $j$'th transverse cavity

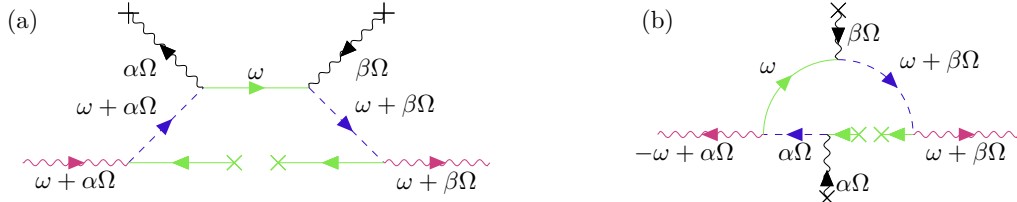

Figure 2: The scattering processes that determine the polarizability of a BEC driven by a PM laser. In these energy-momentum Feynman diagrams, the straight (wavy) lines represent propagation of a bare atom (photon) labeled by its frequency. The momentum/mode indices have been suppressed for brevity. Green solid lines represent atoms in the electronic ground state and blue dashed ones atoms in the excited state. The purple wavy line indicate cavity photons. In addition, there are external sources, represented by lines ending in a cross. Those can be photons from a given sideband of the phase-modulated laser (see discussion in section 2.1), depicted as a black wavy line, or atoms from the BEC, depicted as solid green lines. Panel a) shows the normal process, whereas panel b) shows the anomalous process where the laser acts as a false vacuum.

mode has detuning $\Delta_j = \omega_j - \omega_c$ and is annihilated by the bosonic operator $a_j$. We consider the case where the higher-order cavity modes of interest have an approximate linear energy spacing $\Delta_j \approx \Delta_0 + j\omega_T$. Choosing a PM frequency $\Omega$ that is comparable to $\omega_T$ means that the detuning between higher-order modes and the atomic transition can be compensated by coupling to the higher-frequency pump sidebands, generated by the PM. As the PM is periodic, the sidebands are equidistant in frequency and separated by $\Omega$.

All modes couple to all sidebands but the coupling is inversely proportional to the detuning, $\Delta_j$. Because $\Omega$ is large the only relevant modes are the near-resonant ones

$$\Delta_i \pm \alpha\Omega \sim \Delta_0,\tag{4}$$

where $\alpha$ refers to the laser sideband from eq. (2). The cavity is an intrinsically lossy system, which compensates the energy input from the continuous pumping. This loss is included by coupling the cavity to a continuum of electromagnetic modes playing the role of a bath [43]. At optical frequencies the electromagnetic bath is effectively at zero temperature and is therefore well described with a Markovian loss for the photons. This means that in the von Neumann equation

$$\frac{d\rho}{dt} = -\frac{i}{\hbar}[H,\rho] + D\rho,\tag{5}$$

the unitary Hamiltonian term is supplemented with a Lindblad dissipator [43]

$$D\rho = -\sum_j \kappa_j \left(\frac{1}{2}\left\{a_j^\dagger a_j, \rho\right\} - 2a_j\rho a_j^\dagger\right).\tag{6}$$

This Markovian modeling of the environment is also valid when the PM is included, as long as the environment spectral function appears flat over the energy range $\omega_c \pm \alpha_M\Omega$.

The atom number is assumed to be constant over the duration of the experiment. By coupling the atoms to the cavity the total energy, on the other hand, is not conserved. In the regime where the cavity-mode detunings are comparable to the recoil energy $\epsilon_r = Q^2/(2m)$, where $Q = 2\pi/\lambda_c$ and $\lambda_c$ is the carrier wavelength, this leads to a non-thermal state of the bosonic atoms [44]. In the following, we will neglect this effect by assuming that the time,

$\tau_{\mathrm{rel}}$, required to reach that state is longer than the experimental time, which is realistic for sufficiently large detunings and trap sizes, since $\tau_{\mathrm{rel}}$ scales inversely with these quantities [44]. We will therefore assume the atoms to be initially cooled down to an ultracold temperature $T$ much smaller than the recoil energy, so that thermal excitations are of little concern and we can model the atoms as a perfect BEC.

## 2.2 Polarisability of the Floquet-driven bosonic gas

Collective polaritonic excitations correspond to the normal modes of the electromagnetic field including polarization effects, that is, the modification of the propagation of light due to the medium. In our case, the latter is a gas of ultracold bosonic atoms, whose internal electronic transition is off-resonantly driven by a PM laser, as described in section 2.1. The assumption that the bosons are initially in a perfect BEC at zero temperature corresponds to all atoms being in the electronic ground state and also in the lowest motional state. The motional scales of cold atoms are in the kHz range while the considered PM frequencies are comparable to the TEM mode spacing, which here is considered to be on the order of GHz.

The effect of this driven, polarizable medium on the cavity photons is depicted in Fig. 2, where we illustrate the relevant scattering processes. In Fig. 2 (a), a cavity photon, at frequency $\omega + \alpha\Omega$, excites an atom out of the BEC and takes it to an electronically and motionally excited state (see second line in Eq. (3)). As $\Omega \ll \Delta_a$, the electronic excited state is largely unaffected by the increased frequency, $\alpha\Omega$, of the photon. For the ground state the energy scale is the recoil energy, $\epsilon_r$, which is a much smaller energy than $\Omega$. The scattering process is therefore significantly suppressed if the incoming cavity photon excites the atomic ground state to energies which are large compared to the recoil energy, unless the excess energy is compensated by the laser. With a single laser frequency this is impossible, which leads back to a single-mode scenario. However, via PM the laser sidebands can be brought close to resonance with the high-energy modes. In the formalism this can be efficiently accounted for by splitting the energy ($\omega'$) into Floquet sectors $\omega' = \omega \pm \alpha\Omega$, where $\omega \in \{-\Omega/2, \Omega/2\}$ is the quasienergy. In the diagrams of Fig. 2, this means that if a photon at energy $\omega + \alpha\Omega$ excites the atoms, then the laser has to remove an energy $\alpha\Omega$ for the process to be non-negligible.

The whole sequence is then repeated backwards, leading to the emission of a photon back into the cavity. Due to the modulation, the pump photon used to excite the electronic ground state can be shifted by $\beta\Omega$, which leads to emission into the cavity at this frequency. The same scattering process can clearly also take place with the role of cavity and laser being exchanged, which corresponds to the anomalous process [45, 46] depicted in Fig. 2 (b), where the laser plays the role of a false vacuum.

The crucial point is that, in all these processes, due to the periodic PM of the driving laser, the initial and the final cavity photon can have different frequencies without the atomic ground state propagating at a high energy and thus far off-shell. Due to the absorption and emission of laser photons from different sidebands the polarizability of such a Floquet-driven BEC is thus not diagonal in frequency space as it couples different Floquet sectors of the cavity field differing by multiples of $\Omega$. The modulation removes the penalty of coupling parts of the system modulo $\Omega$ but does not affect momentum conservation.

In the situation considered here, the cavity modes differ in their transverse profile, which to a good approximation is given by a Laguerre-Gauss or Hermite-Gauss function, so that the transverse momentum is not conserved, as we shall discuss below in more detail.

Besides the process illustrated in Fig. 2 (a), the cavities can also couple directly through atoms without any laser. These processes can only couple modes at similar energies (on the scale of recoil energy) and we are considering an experimental regime where the pump power is much higher than the direct atom-cavity coupling. In this case the processes without the laser are significantly suppressed due to the large detuning from the excited state and can therefore

be neglected. Furthermore, one could also consider higher-order processes that would lead to density-density coupling of the cavity. Doing a scaling analysis, like in [45,46], shows that such processes become negligible as one approaches the thermodynamic limit.

Under the condition, stated in section 2.1, that the highest sideband of the laser is still far detuned from the electronic transition, one can adiabatically eliminate the electronic excited state from the diagrams in Fig. 2. This leads to the following dynamical polarizability:

$$\chi_{i,j;\alpha,\beta}(\omega) = \Lambda^2 \bar{c}_\alpha c_\beta \Pi^R_{i,j}(\omega), \tag{7}$$

which has been expressed as a matrix in the TEM-mode basis, with indices $i, j$, and in the Floquet basis, denoted by $\alpha, \beta$. A detailed derivation is given in appendix A. We introduce a single parameter, the effective light-matter coupling strength $\Lambda = \lambda g \sqrt{n_0}/\Delta_a$, which increases linearly with the pump strength and is proportional to the density of the atoms. This can be captured in a single parameter because the energetic difference between the transverse modes is on the order of GHz while the total energy in the electric field is hundreds of THz [47]. This justifies approximating all modes to have a similar coupling to the atoms and, because we explicitly kept the mode functions separated from the coupling strength, the approximation amounts to $g_i = g$.

Again due to the large energy difference between the atomic ground-state motion and the PM frequency $\Omega$ the density response, $\Pi^R$ in Eq. (7), is diagonal in frequency, and takes the form of a bosonic analog of the Lindhard function [48], which specified for a perfect BEC in a trap [45] reads

$$\Pi^R_{i,j}(\omega) = \sum_{n \neq 0} \frac{-2E_n}{(\omega + i0^+)^2 - E_n^2} \times \langle \psi_0 | \eta_j \eta_p | \psi_n \rangle \langle \psi_n | \eta_i \eta_p | \psi_0 \rangle, \tag{8}$$

where $E_n$ is the energy of the atomic eigenmode $|\psi_n\rangle$. We have assumed that the wave functions of the cavity modes, as well as those of the atomic eigenstates, are real. Again, since the transversal modes are not plane waves, momentum is not a good quantum number so the summation over atoms modes is not constrained by momentum conservation.

We see that in this regime the non-diagonal frequency structure of the polarizability, coupling two different Floquet sectors $\alpha, \beta$, is simply encoded in the product of two Fourier coefficients $\bar{c}_\alpha c_\beta$ of the periodic modulation of the pump phase. Each coefficient quantifies how much of the pump intensity goes into the corresponding sideband i.e. how strong that sideband couples to the electronic transition. According to the process shown in Fig. 2, these weights clearly have to enter the polarizability. The remaining part of the polarizability is then given by the retarded density response of the medium, which only depends on the low-energy motional degree of freedom of the atoms in their electronic ground state.

The physical content of the density-response function is quite transparent: it features the matrix element of the transition from the trap ground state $|\psi_0\rangle$ to a motionally excited state $|\psi_n\rangle$, and back to the ground state. The transition out of (back to) the ground state is induced by the spatially-varying optical potential $\eta_i \eta_p$ ($\eta_j \eta_p$), created by the interference between the laser and the cavity field of the $i$'th ($j$'th) TEM mode. As it is clear from inspection of the matrix elements, for a generic choice of the atom trap, the density response, and in turn the polarizability, need not be diagonal in the cavity-mode basis.

In summary, the Floquet-driven bosonic medium can change both the frequency (by multiples of the modulation frequency $\Omega$) and the TEM mode of an incoming photon. Whether and how this happens depends on the specific choice of the PM of the laser, the cavity geometry and the trapping potential for the atoms, as we discuss next.

## 2.3 Symmetric trap and harmonic phase modulation

We consider the radially symmetric configuration of a cigar-shaped harmonic trap in the center of a near-planar cavity. In order to evaluate the density response in Eq. (8), one needs to compute the energy $E_n$ of the state to which the atom is excited out of the condensate and back, as well as the corresponding transition-matrix elements. Assuming a long, cigar-shaped BEC at zero temperature, the longitudinal wave function is well localized in momentum space with a negligible width $\Delta k \ll Q$. Upon absorption of a photon, the atoms therefore scatter into a state with longitudinal momentum $Q$ and recoil energy $\epsilon_r$.

In the transverse direction the atomic state is more complicated due to the presence of the optical lattice formed by the pump standing wave. Assuming a sufficiently weak pump and a transverse trap length much larger than the pump wavelength, we can still approximate the state in the transverse direction to feature again a simple recoil kick, as in the longitudinal direction. We therefore approximate the total energy of the lowest-lying atomic excited state as $E_1 = 2\epsilon_r \equiv E_r$. From here on we will refer to this effective recoil energy, $E_r$, simply as the recoil energy.

Let us now turn to the evaluation of the matrix elements involving the cavity modes. In the longitudinal direction the latter are standing waves and thus momentum conservation is enforced since the atomic states are very close to plane waves as argued above. Instead, in the transverse direction there is no momentum conservation as the cavity modes are Hermite-Gauss or Laguerre-Gauss functions. Still, as the cavity mode function is transversally much broader than the modulation of the atomic states induced by the pump standing wave, we can, to lowest order, approximate the pump mode function as a constant in the matrix element.

The transverse modes which we seek to couple are Laguerre-Gauss (LG) modes, which for the cavity read [49]

$$\mathrm{LG}_{jp}(r, \Theta) = \frac{\mathrm{e}^{-r^2/2w_0^2 + ip\Theta} \sqrt{\frac{j!}{(p+j)!}} \left(\frac{r}{w_0}\right)^{|p|} L_j^{|p|}\left(\frac{r^2}{w_0^2}\right)}{w_0},$$

(9)

with $L_j^{|p|}(x)$ being the associated Laguerre polynomial of order $j$. Due to the approximation of the pump mode function, $\eta_p = 1$, the ground-state atoms have similar transverse mode functions but with $w_0$ replaced by $L_H$.

For atom clouds that are radially symmetric around the cavity axis, angular momentum conservation implies that for each overlap the angular index must be conserved. The BEC is in the ground state with zero angular momentum and therefore only radial cavity modes with the same absolute angular momentum are coupled. Having the carrier frequency being only slightly detuned from the zeroth TEM mode means that only cavity modes with $p = 0$ are relevant and thus $\eta_j(x) = w_0 \mathrm{LG}_{j0}(r)\cos(Qz)$, rendering all mode functions and eigenstates in the density response function real. The remaining radial overlap has a closed form solution shown in appendix B. The simplest case is when the BEC is significantly narrower than the cavity waist. In this case all overlaps are negligible except the one with the atomic state in the zeroth radial mode and a motional state with recoil momentum $E_r$. In this case, the density response takes a simple form

$$\Pi_{i,j}^R(\omega) = \frac{-2E_r}{(\omega + i0^+)^2 - E_r^2},$$

(10)

which is the same as the single-mode result [45]. As the atoms have been integrated away we will from here one refer to the different cavity modes as $\mathrm{LG}_{jp}$ unless otherwise stated.

For non-zero modulation depth it is necessary to parametrize the PM, which can be any real function that is periodic in time. A simple, yet flexible choice is a harmonic modulation:

$$f(t) = B_m \sin(\Omega t),$$

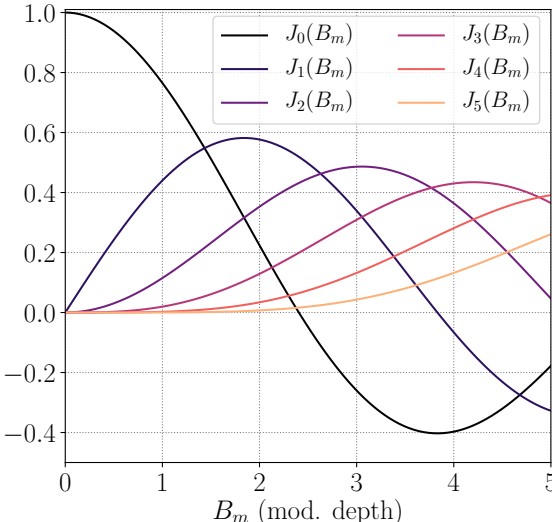

Figure 3: A plot of the first six phase-modulation coefficients as a function of the modulation amplitude. The coefficients are given by the Bessel functions of the first kind.

with the amplitude $B_m$ being a real number. From the Jacobi-Anger expansion [50], the discrete Fourier coefficients are known to be Bessel functions of the first kind $e^{if(t)} = \sum_\alpha c_\alpha e^{i\alpha\Omega t}$, with $c_\alpha = J_\alpha(B_m)$. As shown in Fig. 3, for zero modulation depth, the zeroth-order coefficient is the only non-zero component, which is exactly equivalent to having no PM. Because of the orthonormal nature of the coefficients, the weight in the zeroth-order component is distributed as the modulation is increased. As discussed in section 2.2, since the coefficients directly determine how strongly different modes are coupled by the medium, one can tune the amount of multimodality to a large extent by simply changing the modulation depth.

The modulation frequency $\Omega$ is equally important, as it determines the effective detunings of the different cavity modes. If the modes are exactly linearly spaced one can make the system energetically degenerate by choosing $\Omega$ equal to the mode spacing. If one wants to energetically suppress either higher- or lower-order modes, one can choose a frequency that is either smaller or greater than the mode spacing. Thus, even though the specified harmonic modulation has only two parameters, it is nevertheless highly tunable and has the advantage of being easy to implement experimentally.

## 3 Multimode Floquet Polaritons

Having determined the polarizability of the Floquet-driven BEC, we can now investigate how this modifies the propagation of cavity photons and leads to the formation of polaritons. As already mentioned above, polaritons are the normal modes of the electromagnetic field inside the medium. As such, they appear as poles of the Green's function of the electromagnetic field. In the present case, this Green's function reads

$$\mathcal{D}^R_{i,j;\alpha,\beta}(\omega) = \begin{pmatrix} P^R_{i,j;\alpha,\beta}(\omega) + \chi_{i,j;\alpha,\beta}(\omega) & \chi_{i,j;\alpha,\beta}(\omega) \\ \chi_{i,j;\alpha,\beta}(\omega) & P^A_{i,j;\alpha,\beta}(-\omega) + \chi_{i,j;\alpha,\beta}(\omega) \end{pmatrix}^{-1}, \qquad (11)$$

where the positive- and negative-frequency components of the electromagnetic field have been separated. The resulting Nambu matrix depends on the inverse Green's function of the bare

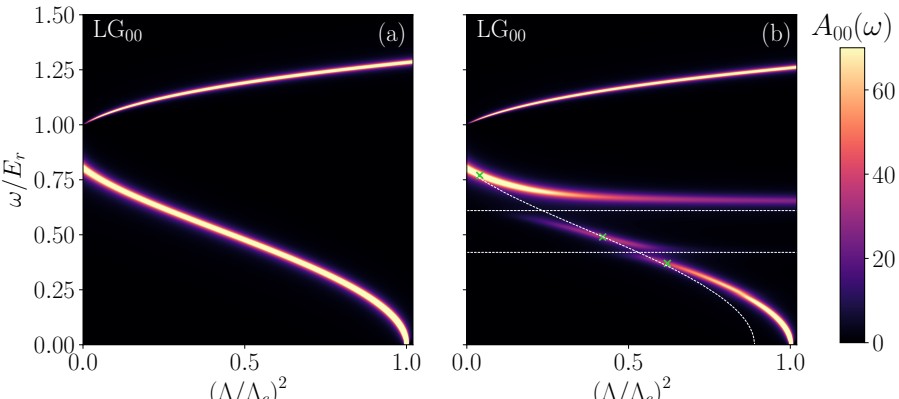

Figure 4: Distribution of the spectral weight in the $LG_{00}$-component of the cavity field. a) when no phase-modulation is applied to the laser, $B_m = 0$ and b) when $B_m = 0.9$, $\Omega = \omega_T + 0.19\, E_r$. The remaining parameters are $\Delta_0 = 0.8E_r$, $w_0/Q = 200$, $L_H = 10^{-3}w_0$ and $\kappa = 0.02E_r$ with the parameters described in section 2.3. The horizontal dashed lines in b) represent the effective detunings of higher-order modes, while the decreasing dashed line represents the position of the spectral line from a). The three green crosses mark the values used in Fig. 5.

cavity

$$P^{R/A}_{i,j;\alpha,\beta}(\omega) = \delta_{i,j}\delta_{\alpha,\beta}\left(\omega - \Delta_j - \alpha\Omega \pm i\kappa_j\right),$$

where $R/A$ indicate the causality of the Green's functions being retarded or advanced. This combination of Green's functions appears here because both positive and negative frequencies are present.

Due to the polarization function in Eq. (11), the Green's function of the electromagnetic field is non-diagonal in both frequency and LG-mode space, such that its computation, in general, becomes rather cumbersome. However, as long as the polarizability decays on an energy scale much smaller than $\Omega \sim \omega_T$, the calculations allow for simplifications. In this case the cavity modes that actually contribute to the electromagnetic Green's function are the ones that are near-resonant modulo a multiple of the modulation frequency: $\Delta_i + \alpha\Omega \sim \Delta_0$. This effectively couples the sideband $\alpha$ with only one mode $i$, largely reducing the number of non-zero elements in Eq. (11).

In order to visualize the poles of the electromagnetic Green's function, we will use the so-called spectral function of the cavity

$$A_{i,j;\alpha,\beta} = i\left(\mathcal{D}^R_{i,j;\alpha,\beta} - \left[\mathcal{D}^R_{j,i;\beta,\alpha}\right]^*\right). \tag{12}$$

This function has peaks in correspondence to the real part of the poles (the polariton frequency), with a width set by the imaginary part (the polariton damping or inverse lifetime). The features observed in the spectral function can be measured by pump-probe [52] or transmission experiments [53].

## 3.1 No phase-modulation

To put our results into perspective and highlight effects of the coupling between many cavity modes, we first review some features of the single-mode calculation [45, 54]. The spectral function for the unmodulated cavity is shown in Fig. 4 (a). When the cavity couples weakly to the atoms the cavity spectrum is dominated by the free Lorentzian peak centered at the detuning $\Delta_0$ with a width determined by the cavity loss $\kappa$. As the pump strength is increased

the atoms start to hybridize with the cavity. This leads to a second peak in the cavity spectral function initially at the recoil frequency, corresponding to the characteristic energy of the density excitation in the atoms. This signature is initially extremely narrow but broadens as the pump strength is increased. In addition, the repulsion between the hybridized modes is also continuously increasing, thereby moving away from the bare resonances. This repulsion between the cavity and atomic peaks signals appreciable hybridization i.e. mixed atom-photon character of the polaritons.

By increasing the coupling $\Lambda$, the low-lying polariton is pushed to lower and lower energies until its frequency reaches zero. At this point, its damping is still finite and the peak in the spectral function is no longer Lorentzian [44, 45]. By further increasing the coupling to a critical value $\Lambda_c$, also the polariton damping vanishes and the normal phase becomes unstable. In the current parameter regime where only the lowest-order atomic transverse mode is relevant, the critical coupling strength $\Lambda_c$ can be found analytically [55]

$$\Lambda_c^2 = \frac{\kappa^2 + \Delta_0^2}{4\Delta_0} E_r \,. \tag{13}$$

If the cavity loss is significantly smaller than the detuning, then $\Lambda_c$ is approximately linearly dependent on $\Delta_0$.

## 3.2 Including phase modulation

Considering the same system, but turning on the PM, additional cavity modes can be brought into play. How strongly these couple to the atoms is determined by the modulation depth according to Eq. (7). As seen in section 2.3, for small atom clouds the density response becomes independent of the mode index, that is the spatial structure of the $LG_{j0}$ cavity modes plays no role in determining how strongly they couple. Therefore, assuming that they have similar detunings (modulo the PM frequency $\Omega$), the cavity mode admixture is fully determined by the PM of the laser. This allows one to create polaritons with a photonic part consisting of a superposition of several cavity modes.

When multiple cavity modes are available, the photon Green's function is a matrix in the cavity-mode basis (see Eq. (11)). As argued above, for the parameters considered here, the matrix structure in the Floquet basis can be suppressed since its index is fixed by the cavity mode, i.e. the Green's function in Eq. (11) is proportional to $\delta_{i\alpha}\delta_{j\beta}$. The spectral function is thus also a matrix in the cavity-mode basis and the four-index spectral function simplifies to a two-index spectral function: $A_{i,j;\alpha,\beta} \to A_{i,j}$. In order to illustrate the effect of the PM, we show the diagonal entry of the spectral function corresponding to the $LG_{00}$ mode in Fig. 4 (b). This allows a direct comparison with the unmodulated case of Fig. 4 (a). Using a PM frequency of the form $\Omega = \omega_T + \epsilon$, the effective detunings of the higher-order modes become

$$\Delta_n = \Delta_0 - n\epsilon \,. \tag{14}$$

Choosing the sign of $\epsilon$ allows one to switch between cases where higher-order modes are effectively lower or higher in frequency than the zeroth mode.

For the spectral function plotted in Fig. 4 (b), $\epsilon$ is chosen to be a small positive number, which causes the higher-order modes to be energetically preferred. With a weak modulation depth ($B_m = 0.9$) the only new relevant modes are $LG_{10}$ and $LG_{20}$, which are given by Eq. (9). The magnitude of the pump sidebands, generated by the PM, is given by the corresponding Fourier coefficient squared. The effective light-matter coupling strength between each mode and the atoms is therefore given by the Fourier coefficient of the nearest sideband (Fig. 3). The relevance of each mode is determined by its atom-coupling strength and the magnitude of the effective detuning. For the specific case of the figure, this means that the $LG_{j0}$ mode

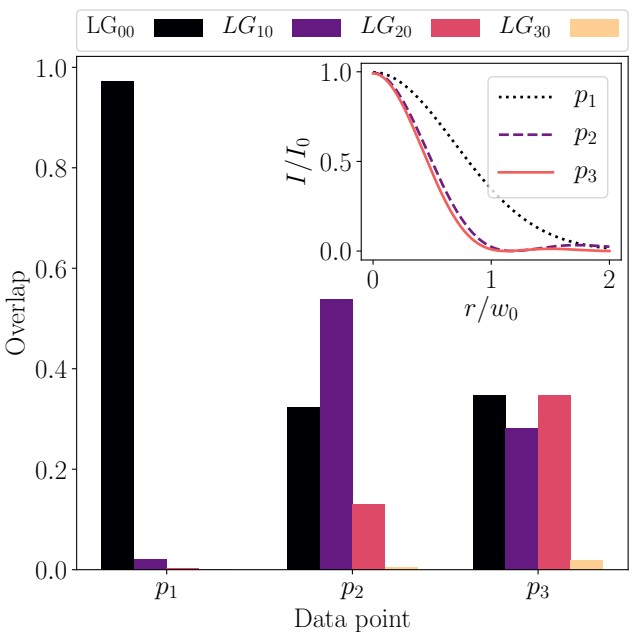

Figure 5: The overlap of different LG modes with the total cavity field at the three points, in $(\omega, \Lambda)$ space, marked by green crosses in Fig. 4 (b): $p_1 = (0.04,\ 0.77)$, $p_2 = (0.42,\ 0.49)$, $p_3 = (0.62,\ 0.37)$. The inset shows the resulting total intensity profiles of the modes in real space, which are radially symmetric. The intensity $I$ is defined as the absolute square of the mode function.

couples only to the $j$'th sideband, which has a magnitude of $J_j(B_m)^2$. Coupling to all the other sidebands leads to far-detuned processes with very small magnitudes.

A first clear signature of the multimode nature is the appearance of more peaks in the spectral function. Looking at Fig. 3 at $B_m = 0.9$ one sees that the zeroth mode has the largest coupling to the atoms. This means that at small $\Lambda$ this mode is the only contribution seen. As $\Lambda$ is increased, $LG_{10}$ starts having a non-negligible coupling to the atoms. Because this mode is $\epsilon$ closer to its sideband than $LG_{00}$ is to the carrier frequency, $LG_{10}$ becomes more favourable for the system and one observes a clear avoided crossing near the frequency of the bare $LG_{10}$-mode. This repeats when the $LG_{20}$ starts being relevant and a second avoided crossing is observed at $\Delta_2$.

These avoided crossings signal the strong hybridization between cavity modes and the emergence of multimode Floquet polaritons. This is confirmed by Fig. 5, where the contributions of the LG modes to the total cavity field are shown for different points on the polariton branches of Fig. 4 (b). This also explicitly demonstrates that the $LG_{30}$ can be neglected as presumed based on its coefficient in Fig. 3.

The avoided crossings allow one to estimate the effective coupling strength between the modes. This is done by fitting the spectrum to that of two linearly coupled modes. The lowest-order procedure is done simply by finding the distance between the two peaks in the spectral function, at an avoided crossing, and dividing it by two. The derivation for this procedure can be found in appendix C. For the first avoided crossing at $(\Lambda/\Lambda_c)^2 = 0.24$, in Fig. 4 (b), the effective coupling strength between the $LG_{00}$ and the $LG_{01}$ mode is $g_{LG_{00},LG_{01}} \approx 3.4\kappa$. The second avoided crossing involves the composite mode, consisting of $LG_{00}$ and $LG_{01}$, and the $LG_{02}$ mode. As the energy of the composite mode is unknown, the exact position of the avoided crossing is also unknown. Furthermore, since three modes are important, the fitting of the

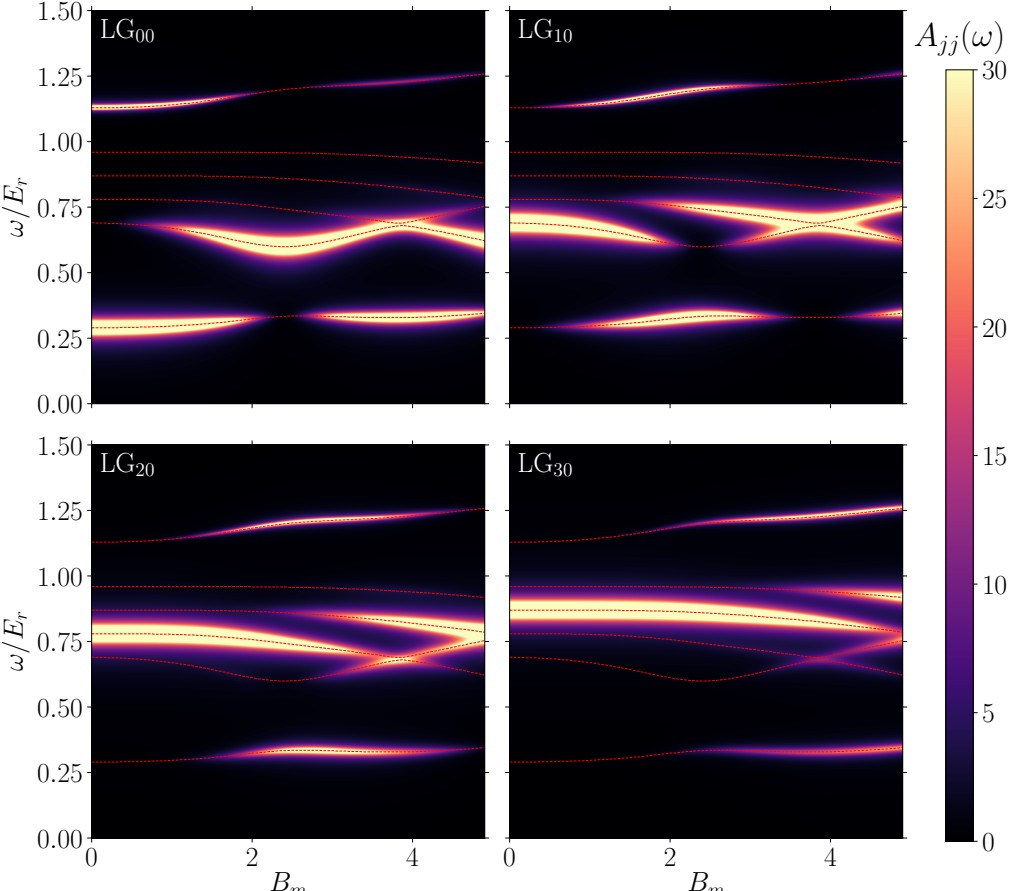

Figure 6: Distribution of the spectral weight as a function of the modulation amplitude for different components of the cavity field, corresponding to the first four radial transverse modes $LG_{pl}$ with $p \in \{0, 1, 2, 3\}$. The PM frequency is chosen such that the higher-order modes have a large effective detuning ($\epsilon = -0.09E_r$) and $\Lambda^2 = 0.7\Lambda_c(B_m)^2$, with the renormalized coupling strength defined in Eq. (15). These figures correspond to cuts through Fig. 4 at $(\Lambda/\Lambda_c)^2 = 0.7$ with a detuning $\Delta_0 = 0.6$ for the zeroth mode, while the remaining parameters are as in Fig. 4. The red lines are poles of the real part of the cavity Green's function in Eq. (11) and therefore indicate the polariton frequencies.

two-mode model is insufficient. This problem can be overcome by investigating the spectral function for the $LG_{02}$ mode ($A_{22}$). In this spectral function only two peaks appear (one for the composite mode and the one for $LG_{02}$) and the effective coupling is proportional to the distance between the two peaks when they have equal height. This avoided crossing is found at $(\Lambda/\Lambda_c)^2 = 0.55$ and the effective coupling strength is $g_{LG_{00,01},LG_{02}} = 1.27\kappa$. The magnitude of the effective couplings has a strong dependence on the amplitude of the sidebands but the exact dependence is non-trivial to extract, as it has to be disentangled from the effects arising from the continuously changing composite nature of the modes. These effects are essential to include, as can be seen from the overlaps at $p_3$ in Fig. 5.

In the inset of Fig. 5 the resulting mode profile of the total cavity field is shown. Already for the few modes involved here, it is clearly seen that the PM leads to a significant decrease of the waist of the cavity field, which is bounded from below by the central waist for the highest involved TEM mode. The observed reduction of the waist of the cavity field directly implies a reduction of the range of the cavity-mediated interactions [29]. More prominent effects can

be achieved by increasing the PM amplitude, but also by displacing the atom cloud from the center of the cavity. The oscillating nature of higher-order transverse modes can then lead to interference effects that further decrease the interaction range.

Changing the modulation depth $B_m$ gives the freedom to choose how strongly the different modes couple to the atoms. The parameters $B_m$ and $\epsilon$ are independent and therefore allow one to tune between a wide range of different multimode scenarios. In Fig. 4 (b) we showed how choosing $\epsilon > 0$ leads to avoided crossings due to the higher-order modes being energetically favoured. In Fig. 6 we choose $\epsilon < 0$ which means the zeroth mode is always energetically most preferable. The ratio $\Lambda/\Lambda_c = 0.7$ is kept fixed for all modulation amplitudes, while the latter is varied over a range where only the first four modes are relevant. In order to facilitate the visualization of the avoided crossings, the approximate polariton frequencies are overlaid as red-dashed lines.

Consider first the case of no modulation ($B_m = 0$). The $\text{LG}_{00}$ mode's spectral function shows the familiar two peak structure discussed in Fig. 4 (a), whereas the higher-order modes have no atom peak in the spectrum but only their bare Lorentzian lineshape at the respective detunings. As $B_m$ is increased the other modes start to couple through the atoms at the price of decreasing the atom coupling of the $\text{LG}_{00}$ mode. This is seen by the fact that the $\text{LG}_{10}$ line starts showing up in $A_{00}$. As seen in Fig. 3, the coefficient $c_0$ vanishes near $B_m \approx 2.3$. This means that the $\text{LG}_{00}$ mode no longer couples to the atoms, and both the atom peak and what will become the superradiant peak vanish from $A_{00}$. At that modulation depth, $A_{00}$ is simply the free Lorentzian line shape at $\Delta_0$. Meanwhile, both $\text{LG}_{10}$ and $\text{LG}_{20}$ couple relatively strongly to the atoms, and therefore effectively to each other, which introduces multiple peaks in the corresponding components of the spectral function. Further increasing $B_m$ reintroduces the $\text{LG}_{00}$ mode in the polariton peaks.

This shows that one can tune $B_m$ such that the polaritons have contributions from many modes. In particular, the polariton becoming unstable at the superradiant transition will then be a linear combination of all modes $\text{LG}_{j0}$ with $j < \alpha_M$, with weights given by histograms similar to Fig. 5. As the higher modes have a larger detuning than the zeroth mode, it is necessary to increase the pump power in order to keep the ratio $\Lambda/\Lambda_c$ fixed. This shows up in the spectral functions, where the atomic peak is pushed to higher energies as $B_m$ is increased. It is interesting to note that, even though the higher-energy modes are red detuned, one can choose a modulation depth which results in anti-crossings, especially prominent at around $B_m = 4$. This demonstrates that non-trivial multimode effects can be seen using both $\epsilon < 0$ and $\epsilon > 0$.

## 4 Multimode superradiance

Having multiple cavity modes available affects several features of the transition to the superradiant phase. The first clear effect is that the hybridization between multiple cavity modes can give rise to an increase in the critical coupling $\Lambda_c$ compared to the unmodulated system, as seen in Fig. 6 by the atom peak being pushed to higher energies. The behaviour of $\Lambda_c$ is plotted as $B_m$ is changed in Fig. 7 (a). To isolate the effect, consider first the case where $\Omega = \omega_T$. In this case all modes in the system have the same effective detuning. The value of $\Lambda_c$ one would observe in an experiment is the black dashed line increasing non-monotonically as the PM is increased. This rise in the critical coupling strength is due to the PM being symmetric around the carrier frequency. The higher-order modes see only the closest laser sideband, but the PM also induces sidebands at a lower energy than the carrier frequency. For these negative frequency sidebands the cavity has no stable modes as they correspond to very high transverse quantum numbers with a lower longitudinal quantum number. These modes are lossy due

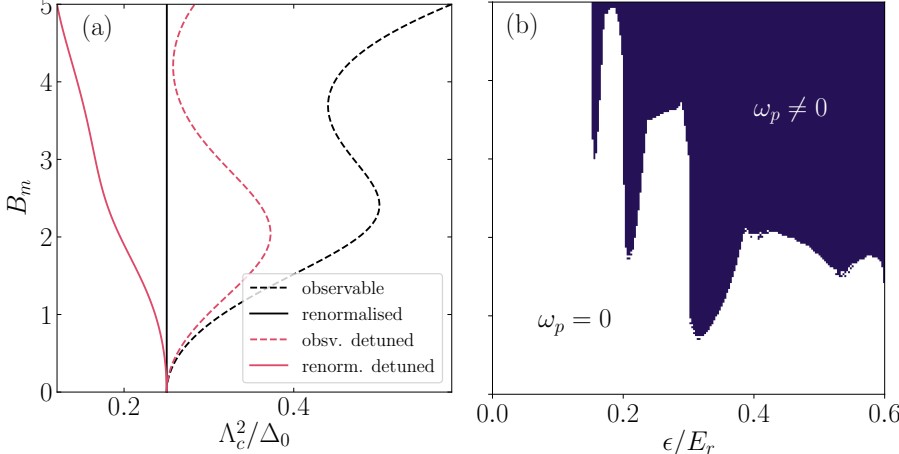

Figure 7: a) A plot of how the critical coupling strength $\Lambda_c$ changes with the modulation amplitude. The two different colors represent $\Omega = \omega_T$ (black) and an effective detuning between the modes $\Omega = \omega_T + 0.1$ (red). The remaining parameters are the same as the ones used in Fig. 6. The dashed lines are obtained for the same systems but with the critical coupling strength being renormalized to make up for the lost power in the unused laser sidebands given by Eq. (15). b) Plot of the dominant instability of the system as a function of the PM parameters with the same parameters as in Fig. 6 but with $\kappa = 0.05\, E_r$. In the white region the system experiences a transition to a superradiant phase at the finite critical coupling strength $\Lambda_c$. The colored region indicates parameters where a finite-frequency instability takes place.

to their large transverse size and are generally not stable. The power in the lower frequency peaks is therefore effectively lost. This means that the total effective pumping power of the cavity is decreased by the weight in the negative frequency coefficients. This effect can be taken into account by renormalising the coupling

$$\Lambda^2 \to \frac{\Lambda^2}{\sum_{0 \le \alpha < \alpha_M} c_\alpha^2} = \Lambda(B_m)^2. \tag{15}$$

We always use the explicit $B_m$ dependence to denote the renormalized coupling strength. Employing this renormalization one sees that $\Lambda_c(B_m)$ stays constant through all modulation depths. When $\epsilon > 0$, as in Fig. 4 (b), the higher-order modes have a smaller detuning, and one again sees that $\Lambda_c$ is increasing with $B_m$. This might seem counter-intuitive, as the higher modes have smaller effective detunings and one would therefore expect $\Lambda_c$ to decrease with $B_m$, which is indeed observed when using the renormalized coupling strength. We note that the loss of pump power due to unused sidebands can be avoided in a simple manner by placing the carrier frequency close to a higher-order cavity mode.

Another important modification to single-mode superradiance is that with a positive $\epsilon$, higher-order modes have a lower frequency than the zeroth mode. For a range of values of $\epsilon$, $B_m$ and $\Delta_0$, a situation arises where some higher-order modes are effectively red detuned from their nearest sideband. In this case the magnitudes of their detunings are still small, compared to $\omega_T$, but now have negative signs. For a single-mode system, this leads to an instability at a much smaller $\Lambda_c$ than for a similarly blue-detuned cavity mode. The critical coupling is much smaller because the bare atomic part of the system has a vanishingly small loss and once the cavity is red detuned the atomic part of the system is easily rendered unstable, even at very weak interaction strengths. This type of "atomic instability" happens at finite frequency, that is, the atomic polariton becoming unstable still has a finite energy. This

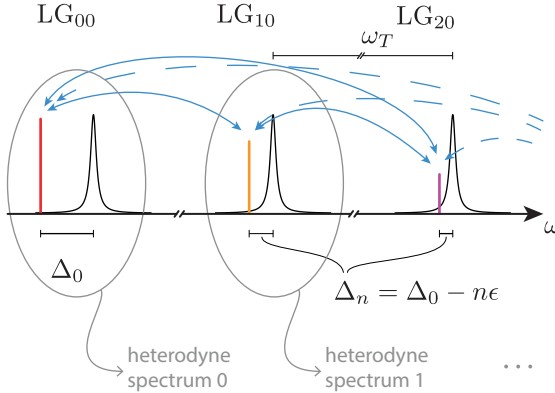

Figure 8: Modulation scheme for $\epsilon > 0$. The detuning $\Delta_n$ between higher-order modes (lineshapes indicated by the Lorentzian peaks spaced by $\omega_T$) becomes increasingly smaller. For weak PM, the amplitudes of the sidebands (colored lines) decrease. The blue arrows indicate the coupling between the different cavity modes due to the interaction with the atomic system. The spectra in the vicinity of the different modes of the cavity can be conveniently separated with a heterodyne detection system by tuning the frequency of the local oscillator close to a cavity mode frequency and electronically filtering out the contributions from the modes. This results in several heterodyne spectra as indicated by the grey ellipses.

instability is thus of a very different nature than the superradiant transition happening at $\Lambda_c$, via a zero frequency excitation.

The results presented so far assume that the system remains stable up to $\Lambda_c$, which means that we have to choose the parameters such that the finite-frequency instability is avoided. Fig. 7 (b) shows the nature of the leading instability as a function of the PM parameters. For $\epsilon < 0.15 E_r$ the transition always happens to a stationary, superradiant phase. This is explained by the fact that we have considered a system with 5 available modes and with $\Delta_0 = 0.6$. This means that the smallest detuning for this region is $\Delta_4 = \Delta_0 - 4\epsilon \geq 0$, which does not allow for a finite-frequency instability. Clearly, the constraint on the number of cavity modes imposed by the physical setup plays an important role for the existence of the finite frequency instability. When moving to larger values of $\epsilon$ it can be seen that, due to the effective interactions between multimode polaritons, the boundary between zero- and finite-frequency instabilities is highly non-linear and shows the, for dynamical systems, often occurring Arnold tongues. For the parameters considered in the present work, there is thus a large region where the PM can be used to generate multimode Floquet polaritons without encountering a finite-frequency instability.

## 5 Experimental considerations

### 5.1 Proposed realizations and observability

The scheme proposed here can be realized in several state-of-the-art laboratories. The main feature we discussed is the presence of anti-crossings, which is a clear signature of the formation of the multimode Floquet polaritons. A prerequisite for achieving appreciable hybridization between different cavity modes is that two resonances in the spectral function have to be brought close to each other. As all cavity modes have a positive detuning, the corresponding peaks move towards zero frequency when coupling to the atoms. In order to generate an

(avoided) crossing, we therefore need to make a mode of higher energy move to zero faster than a lower-energy mode, while they both couple to the atoms.

To achieve this there are two different strategies that can be realized with different parameter choices of the PM: First, by having $\epsilon > 0$ there is a higher-order mode that has a smaller detuning. This situation is shown in Fig. 8, and the resulting spectrum was displayed in Fig. 4 (b). By introducing a shallow PM, the zeroth order cavity mode will couple most strongly to the atoms. This will push it faster towards zero than the higher-order mode which couples weaker to the atoms. As the zeroth mode comes close to the higher-order cavity mode, their repulsion will be determined by how strongly they respectively interact through the atoms. This is a good strategy because the highest energy mode ($LG_{00}$) is, for shallow modulation depth, more strongly coupled to the atoms and can therefore lead to a large intersection angle with the lower energy modes. Nevertheless, one could also use $\epsilon < 0$ in combination with a deeper modulation, as described in Fig. 6. Generating the strongest hybridization between the cavity modes is therefore a question of choosing the parameters such that the higher-energy mode moves faster but the lower-energy mode still couples significantly to the atoms. Both approaches allow for hybridization of the different modes but work in different regimes. The choice of the ideal approach thus depends on the specific realization.

A general requirement to observe the mode crossing is to have cavity modes with detunings smaller than the effective recoil energy of the atoms. This is a significant constraint for the experimental observability, as it means that the cavity loss rate has to be smaller than the effective recoil energy. The weaker the losses, the sharper the avoided crossings that can be observed. Loss rates below the recoil energy can be obtained in long cavities (several centimetres) given that the current technologies only allow for a finesse of the order of typically $5 \cdot 10^5$ in optical cavities. Nevertheless, by choosing geometries with tight focuses (near confocal configurations), strong collective atom-light interaction can still be achieved, as demonstrated in an experiment using $^{87}$Rb atoms [21,23,56]. Here the mode diameter is small at the position of the atom cloud and the loss is kept small because of the long round trip distance of the cavity photon. It is worth mentioning that due to the mass dependence of the recoil energy, the numbers are more favourable for lighter atoms such as Lithium. Recoil resolution here can be reached in cavities with larger line widths [57], which reduces the technical challenges.

## 5.2 Modulation schemes

The pump modulation scheme discussed so far assumed a PM of the carrier frequency of a laser, e.g. by driving an electro-optical modulator with a single frequency. Such a PM of a laser field generates upper and (in the discussed scheme unused) lower sidebands at integer multiples of the modulation frequency and with relative weights dictated by the Bessel function, see Fig. 3. This restricts the possible effective photon couplings that can be induced with the scheme described, but nevertheless can address dozens of modes if one reaches high enough PM depths with the EOM. Many more modes can be accessed using optical frequency combs as they can be generated using two-stage modulation schemes [40].

The set of realizable interactions can be significantly extended if the electro-optical modulator is driven by multiple frequencies in the weak modulation regime, where only the $J_1$ Bessel function, which describes the spectral weight of the first sideband, is considerably populated and grows monotonously in modulation depth. This way, each tone creates its own sideband with arbitrary coupling that is set by its individual amplitude, phase, and frequency which are all determined by the synthesized signal of e.g. an arbitrary waveform generator or with software defined radio techniques. This approach would extend our proposed scheme to hundreds of transverse modes, each coupled with tailored coupling strength.

Reaching this high numbers of modes would allow to appreciably reduce the cavity-mediated interaction range below the cavity waist, making the study of models with

finite-range interactions possible. Moreover, the additional tailoring of the coupling to each cavity mode separately would allow to go beyond an interaction potential which monotonously decays in space. From a fundamental perspective, this prompts the investigation of strong frustration effects and further exotic types of spatial correlations. From a technological perspective, such a flexibility could for instance enhance the efficiency of quantum annealing protocols [58].

## 5.3  Detection methods

The interactions between the induced multimode Floquet polaritons can be experimentally characterized in different ways. The light field leaking from the cavity will carry information both in its spatial profile and in its frequency spectrum. Different from the situation of a fully degenerate multimode cavity [28], the contribution of the individual modes can be analysed in frequency space, as is indicated in Fig. 8. The light field leaking from the cavity is sent to a heterodyne detection scheme, where the frequency of the local oscillator is tuned to be close to the cavity mode of interest, such that the effect from all other modes can be filtered. The heterodyne detection then allows to access the spectrum of the field in vicinity of that mode. The required frequency separation between the cavity modes ($\sim \omega_T$) and the spectral features ($\sim E_r$) is naturally given in the proposed scheme.

Clear multimodality should also be observable by decomposing the cavity output light after entering the superradiant phase, as has been demonstrated for degenerate cavities [28]. Differently from the avoided crossings, a multimodal cavity output should be visible even when the loss rate is much larger than the effective recoil energy. Additionally, absorption images of the atomic cloud after ballistic expansion from the superradiant phase will allow to characterize the atomic mode composition of the polaritons.

## 6  Conclusion

We have shown that the periodic phase modulation of the driving laser can generate a large dispersive coupling between an ultracold atomic cloud and many modes of a non-degenerate cavity. This leads to the formation of multimode Floquet polaritons. Their mutual interactions mediated by the atoms are visible as avoided crossings and ultimately lead to a phase transition to a multimode superradiant state. This scenario should be experimentally testable in state-of-the-art platforms.

In this investigation, we have focused on the energetic effect of the PM and simplified the degrees of freedom by considering small atom clouds at the center of the cavity. Having seen that multimode Floquet polaritons can be generated, a very interesting aspect is to explore regimes where the cloud has significantly different overlaps with the different resonant modes. This allows one to establish an added competing effect that can further enrich the multimode correlations. Moreover, the multimode nature of the polaritons and their mutual interactions might give rise to a richer scenario for finite-frequency instabilities. This adds significant structure to the ordered phase, which gives rise to qualitative features that can differ from the known single-mode case. We defer these studies to future work.

## Acknowledgements

We thank Davide Dreon, and Tilman Esslinger for useful discussions. T.D. acknowledges funding from SNF Project No. IZBRZ2 186312 and the NCCR QSIT.

# A  Derivation of dynamical polarizability

In this appendix, we will derive the polarization function of the Floquet driven bosonic gas given in Eq. (7) and Eq. (8), which is used in the main text to compute the cavity spectrum. Because the cavity has losses, it is important to treat both the unitary and non-unitary evolution of the system on the same footing. To this extent we use a path integral formulation of the time evolution of the system on the Keldysh contour. The Keldysh real-time contour is composed of a backward and a forward piece, which allows to treat situations away from thermal equilibrium and in particular open systems whose state is described by a density matrix. In this path-integral formulation, each field corresponding to a given degree of freedom acquires a further index with two possible values, one for each piece of the time contour. Here we rotate these two additional indices by performing the so-called Keldysh rotation [51], after which we deal with a "classical" and a "quantum" component of the field. For bosons, the classical field can be thought of as the field that can acquire a finite expectation value, whereas the quantum field describes fluctuations around this value. Due to the doubling of the fields, the theory also has two independent Green's function: the retarded (or advanced) and the Keldysh Green's function. The retarded Green's function only contains spectral information about the excitations of the system, while the Keldysh Green's function also encodes the occupation of these excitations. All the results discussed in this paper are solely relying on the spectral properties. The Lindblad master equation, Eq. (5), is used as the starting point for constructing the action [59].

A main advantage of the Keldysh path-integral formulation is that it allows to apply all the standard field-theory techniques available also in thermal equilibrium. Here we make use in particular of Feynman diagrams to organise and interpret the different contributions to the photon self-energy. The latter describes the dressing of the photon via the interaction with the atomic medium, that is, the polarization function of the medium. The periodic phase-modulation of the laser dressing the atoms complicates the energy structure of the scattering between cavity photons and atoms, and the diagrammatic formulation turns out to be extremely useful for the understanding. To construct Feynman diagrams one needs to construct the non-equilibrium action from Eq. (5). The free action takes the form

$$
\begin{aligned}
S_0 = \int d\mathbf{x}\, d\mathbf{x}' & \left[ \begin{pmatrix} \bar{\psi}_c \\ \bar{\psi}_q \end{pmatrix}_{\mathbf{x}}^T \begin{pmatrix} 0 & G_\psi^{A^{-1}} \\ G_\psi^{R^{-1}} & P_\psi^K \end{pmatrix}_{\mathbf{x},\mathbf{x}'} \begin{pmatrix} \psi_c \\ \psi_q \end{pmatrix}_{\mathbf{x}'} + \begin{pmatrix} \bar{\phi}_c \\ \bar{\phi}_q \end{pmatrix}_{\mathbf{x}}^T \begin{pmatrix} 0 & G_\phi^{A^{-1}} \\ G_\phi^{R^{-1}} & P_\phi^K \end{pmatrix}_{\mathbf{x},\mathbf{x}'} \begin{pmatrix} \phi_c \\ \phi_q \end{pmatrix}_{\mathbf{x}'} \right] \\
& + \int dt\, dt' \sum_i \begin{pmatrix} \bar{a}_{i,c} \\ \bar{a}_{i,q} \end{pmatrix}_t^T \begin{pmatrix} 0 & D_i^{A^{-1}} \\ D_i^{R^{-1}} & P_{a,i}^K \end{pmatrix}_{t,t'} \begin{pmatrix} a_{i,c} \\ a_{i,q} \end{pmatrix}_{t'},
\end{aligned}
\tag{16}
$$

where $\mathbf{x} = (t, \mathbf{r})$ denotes the full space-time coordinate and $c/q$ are the previously mentioned "classical" and "quantum" indices, denoting how the fields are distributed on the Keldysh contour [51]. Here we follow the notation used for the Hamiltonian in Eq. (3).

The non-interacting inverse Green's functions for the atoms are given by

$$
\begin{aligned}
\left( G_\psi^{R/A} \right)^{-1}(\mathbf{x}, \mathbf{x}') &= \delta(\mathbf{x} - \mathbf{x}') \left( i\partial_{t'} + \frac{\nabla^2}{2m} - V_g(\mathbf{r}') + \mu_\psi \pm i0^+ \right), \\
P_\psi^K(\mathbf{x}, \mathbf{x}') &= i\, 2\, \delta(\mathbf{x} - \mathbf{x}') F_\psi(\mathbf{x}') 0^+, \\
\left( G_\phi^{R/A} \right)^{-1}(\mathbf{x}, \mathbf{x}') &= \delta(\mathbf{x} - \mathbf{x}') \left( i\partial_{t'} - \Delta_a \pm i0^+ \right), \\
P_\phi^K(\mathbf{x}, \mathbf{x}') &= i\, 2\, \delta(\mathbf{x} - \mathbf{x}') 0^+.
\end{aligned}
\tag{17}
$$

$F_\psi$ is the energy distribution function of the atoms in their internal ground state, which in Fourier space reads $F_\psi(\omega, \mathbf{k}) = \coth\left( (\omega - E_\mathbf{k})/2T \right)$. $\mu_\psi$ is the chemical potential and since

we are considering thermal bosons, it must always be smaller than or equal to zero [48]. As $\Delta_a$ is the largest energy scale, the excited state is only virtually occupied and its distribution function is therefore 1 [51]. The components of the inverse cavity Green's function are given by

$$\left(D_i^{R/A}\right)^{-1}(t,t') = \delta(t-t')(i\partial_{t'} - \Delta_i \pm i\kappa_i)\,, \tag{18}$$
$$P_{a,i}^K(t,t') = i\,2\,\delta(t-t')\kappa_i\,.$$

When expressed in the classical/quantum basis, the interaction part of the action is found to be

$$S_I = -\lambda \int d\mathbf{x}\left(e^{-if(t)}\begin{pmatrix}\bar{\phi}_c \\ \bar{\phi}_q\end{pmatrix}_{\mathbf{x}}^T \sigma_1 \eta_p(\mathbf{r})\begin{pmatrix}\psi_c \\ \psi_q\end{pmatrix}_{\mathbf{x}} + H.c.\right) \tag{19}$$

$$-\sum_i \frac{g_i}{\sqrt{2}} \int d\mathbf{x}\left(\eta_i(\mathbf{r})\begin{pmatrix}\bar{\phi}_q\psi_c + \bar{\phi}_c\phi_q \\ \bar{\phi}_q\psi_q + \bar{\phi}_c\psi_c\end{pmatrix}^T \begin{pmatrix}a_{i,c} \\ a_{i,q}\end{pmatrix} + H.c.\right). \tag{20}$$

We are only considering the non-superradiant phase which means that, the cavity is on average empty so that there is no additional external potential for the atoms. One can therefore use the appropriate eigenbasis (depending on the trap) to diagonalise the spatial part of problem. In the dispersive regime, the spatial features of the excited internal state can be neglected, so any orthonormal basis can be used. As described in the main text, we consider a perfect BEC at zero temperature for the ground-state atoms. To include the macroscopically occupied ground state the "classical" field is written as $\psi_c = \xi_0 + \sum_{n'}\psi_{c,n'}$, with $\xi_0^2 = n_0$ being the density of the condensate and $\psi_{c,n'}$ the non-condensate component of the classical field in eigenstate $n'$ with $n' \neq 0$. We are neglecting fluctuations of the condensate itself, which means that the quantum field only has non-condensate components $\psi_q = \sum_{'n}\psi_{q,n'}$. Because the driving is periodic, the system exhibits a discrete time-translation invariance which makes the physical interpretation simpler after Fourier transforming the action. In Fourier space the interaction has the form

$$S_I = -\sum_{m,\alpha}\lambda\int\frac{d\omega\,d\epsilon}{2\pi}c_\alpha\left(\sum_{n'}\langle\phi_m|\eta_p|\psi_{n'}\rangle\begin{pmatrix}\bar{\phi}_{c,m} \\ \bar{\phi}_{q,m}\end{pmatrix}_\omega^T \sigma_1\begin{pmatrix}\psi_{c,n'} \\ \psi_{q,n'}\end{pmatrix}_\epsilon \delta(\omega-\alpha\Omega-\epsilon)\right.$$

$$\left.+\langle\phi_m|n_p|\psi_0\rangle\bar{\phi}_{q,m}\xi_0\delta(\omega-\alpha\Omega)+H.c.\right)$$

$$-\sum_{i,m}\frac{g_i}{\sqrt{2}}\int\frac{d\omega\,d\epsilon\,d\rho}{(2\pi)^2}\left(\sum_{n'}\langle\phi_m|\eta_i|\psi_{n'}\rangle a_{\beta,i}\psi_{\gamma,n'}\bar{\phi}_{\nu,m}M^{\beta,\gamma,\nu}\delta(\omega-\epsilon-\rho)\right. \tag{21}$$

$$\left.+\langle\phi_m|\eta_i|\psi_0\rangle\begin{pmatrix}\bar{\phi}_{c,m} \\ \bar{\phi}_{q,m}\end{pmatrix}_\omega^T \sigma_1\begin{pmatrix}a_{c,i} \\ a_{q,i}\end{pmatrix}_\epsilon \xi_0\delta(\omega-\epsilon)+H.c.\right),$$

where $n'/m$ labels the non-condensate component of internal ground/excited state of the atoms, and $\alpha$ stems from the Fourier series representation of the drive modulation in Eq. (2). The Einstein summation notation has been used to simplify the Keldysh index combinations with $M^{\beta,\gamma,\nu} = (\sigma_1^{\gamma,\nu}, \mathbb{1}^{\gamma,\nu})^\beta$, with the classical (quantum) index being the first (second) position. The free Green's functions are easily Fourier transformed as they are fully time-translation invariant but it is instructive to first inspect the interaction more thoroughly. As discussed in the main text, the periodic nature of the drive makes it convenient to fold everything into one energy interval of width $\Omega$. Because of the similarity to the Bloch construction for spatially periodic potentials, this first energy interval is commonly referred to as the 1st Floquet Brillouin zone (1FBZ). We choose the 1FBZ to be symmetric around 0 such that the quasi-energy is $\omega = \{-\frac{\Omega}{2}, \frac{\Omega}{2}\}$.

The condensate is described as a classical field with a macroscopic value. It is featured in the spectral function as a delta peak at zero energy and is only present in the zero'th energy band at quasi-energy 0. Here it is macroscopically occupied with a density of $n_0$. The non-condensate component is unoccupied, but contributes to the spectral content with peaks appearing in correspondence to all non-zero motional eigenstates of the atoms in their electronic ground state. Even though these peaks sit at non-zero energies, the motion of the atoms is still much slower than the PM frequency. The non-condensate peaks of atoms in internal ground state are therefore also only present in the zero'th energy band. This is in contrast to the internal excited state, which has spectral weight at the large energy scale $\Delta_a$. As this scale is by far the largest in the problem, and in particular much larger than the modulation frequency, we approximate the spectral weight of the internal excited state of the atom to be the same in all energy bands.

Finally, each cavity mode has a different weights in each energy bands, but the constraint described by Eq. (4) is such that a given mode effectively contributes to just a single band. In summary, the non-interacting Green's functions for the unoccupied degrees of freedom in the 1FBZ are then given by

$$
\begin{aligned}
G_{\psi,\alpha,n'}^{R/A}(\omega) &= \delta_{\alpha,0}\left(\omega - E_{n'} \pm i0^+\right)^{-1}, \\
G_{\psi,\alpha,n'}^{K}(\omega) &= G_{\psi,\alpha,n'}^{R}(\omega) - G_{\psi,\alpha,n'}^{A}(\omega), \\
G_{\phi,\alpha,m}^{R/A}(\omega) &= \left(\Delta_a^{-1} \pm i0^+\right)^{-1}, \\
G_{\phi,\alpha,m}^{K}(\omega) &= G_{\phi,\alpha,m}^{R}(\omega) - G_{\phi,\alpha,m}^{A}(\omega), \\
\left(D_{i,\alpha}^{R/A}\right)(\omega) &= \left(\omega - \Delta_i - \alpha\Omega \pm i\kappa_i\right)^{-1}, \\
P_{a,i,\alpha}^{K}(\omega) &= i2\kappa_i.
\end{aligned}
\tag{22}
$$

The chemical potential of the ground-state atoms is set to zero so that the energy of the ideal BEC identically vanishes.

Given the interaction in Eq. (21), we construct all the contributions to the cavity-photon self-energies up to second order in the cavity-atom coupling strength $g_i$. The self-energy is most simply represented using diagrams in the energy representation like the ones shown in Fig. 2. In this representation, the direction of the arrows indicates which way the energy (given by the label next to the line) is flowing. Because the laser is included in the diagrams, the vertices are energy conserving. The self-energy is equal to the polarizability except for a different sign convention. Using the self-energies one can solve the corresponding Dyson equations, as in Eq. (11), which is a resummation to infinite order of the chosen perturbative processes, yielding the photon Green's function.

The diagrams are used to illustrate the physical processes, so for conciseness we do not include the the Keldysh structure in the diagrams. The Keldysh structure leads to a large number of similar diagrams with the same topology many of which can be related and cancel due to causality [51]. This structure is important to find the actual value of the self-energies but not to understand the physical processes considered here.

The first process that emerges, due to the adiabatic elimination of the excited state, is the Stark shift diagram shown in Fig. 9 (a). This process virtually excites the ground state by coupling to the laser and subsequently decays back to the ground state by emitting into the laser. The cavity is not involved in the process but the process renormalises the ground-state atoms which does affect the cavity coupling. Because the ground-state atoms move slowly compared to the PM frequency the ground-state atoms only sees the time-averaged Stark shift. Due to the normalisation of the PM the average is equivalent to the unmodulated Stark shift. The shift, which is of order $\mathcal{O}(\lambda^2/\Delta_a)$, can then be absorbed into the chemical potential of the ground-state atoms. This new effective chemical potential contains both the Stark shift, the

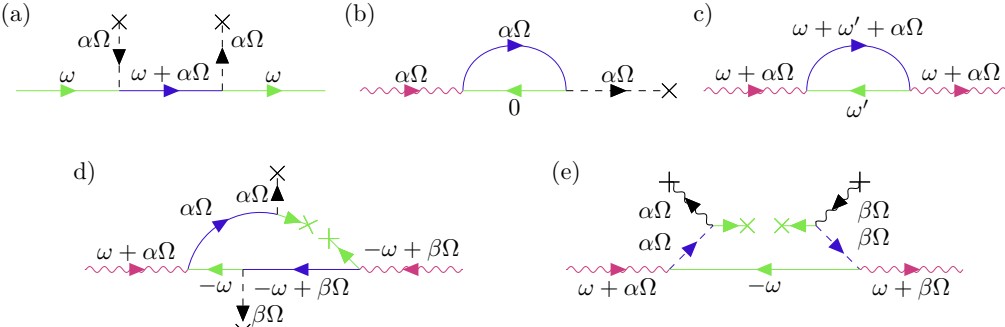

Figure 9: All the "geometrically" different Feynman diagrams describing the different processes in the system up to $\mathcal{O}(g^2)$. The notation is equivalent to the one used for the two diagrams in the Fig. 2. a) Due to the coupling to the excited-state the ground-state experiences a Stark shift. The pump excites the ground-state and the excited-state then decays by emitting into the pump. b) The process that is first order in cavity coupling. The cavity excites the atom which then decays into the laser. The opposite process also exists. c) The two-photon process which is second order in cavity coupling. A photon excites the atom and the excited state subsequently decays into the same cavity mode. This process is not amplified by the laser. d) An example of the opposite anomalous process shown in Fig. 2 (b). e) The opposite process of Fig. 2 (a) that gives the second term in the self-energy/polarizability.

vacuum fluctuations and the chemical potential itself. Because we are considering the atom to be in a perfect BEC this effective chemical potential is forced to zero and the ground state Green's function is therefore effectively unmodified by the Stark shift.

The first type of processes that involves the cavity are of order $\mathcal{O}(g_i/\Delta_a)$ and shown in Fig. 9 (b). In this process a cavity excites the atom which then decays into the laser. This type of processes vanishes because we assume a homogeneous atom cloud (non-superradiant phase and large longitudinal trap). A homogeneous cloud leads to purely destructive interference among the scattered photons so there is effectively no Rayleigh scattering. In the equations this manifests by the longitudinal spatial integral evaluating to zero.

The next process is of order $\mathcal{O}(|g_i|^2/\Delta_a)$ and is the two photon process without the laser, shown in Fig. 9 (c). Because the laser is not partaking in this process, it does not lead to any coupling between higher-order cavity modes. As the cavity-atom coupling is a small parameter while $\Delta_a$ is huge, there is a large regime where these processes can be neglected compared to the process that are amplified by the laser driving.

If the laser is involved in the scattering then the processes shown in Fig. 9 (e) and 2 (a) emerge. This is of order $\mathcal{O}(\lambda^2 g_i \bar{g}_j/\Delta_a^2)$. This type of process can be made important by increasing the laser strength to make up for the small cavity-coupling strength and the large atom detuning. In this regime the PM can be used to push the higher-order cavity modes closer to the bare detuning and thereby making them energetically relevant. In this scattering process a cavity photon is first annihilated in one mode and then created again at another (or the same). We call this a normal process. This is in contrast to the process shown in Fig. 9 (d) and Fig. 2 (b) where two cavity photons are either annihilated or created by using the laser as a false vacuum. We use the nomenclature from superconductivity and denote these processes as the anomalous scattering processes. The anomalous processes have the same value as the normal process and are therefore as important. To include them we rewrite the action in a Nambu form [48] which directly leads to the matrix structure of the retarded Green's function in Eq. (11).

Both of the above relevant scattering processes are affected by our assumption of all initial

occupation being in the BEC. This means that all diagrams that only involve thermal ground states cancel out and the only remaining diagrams are ones of similar topology but with one of the thermal ground states being replaced by the condensate as is the case in Fig. 9 (d), (e) and Fig. 2 (a), (b). Because of the BEC coupling, all diagrams where the BEC appears, are enhanced by the atom density. This makes the self-energies important within laser powers easily reachable by experiments.

The retarded cavity self-energy describes how the free cavity spectrum is affected by the coupling to the atoms. Adiabatically eliminating the excited state, the normal part of the retarded self-energy is given by the sum of Fig. 9 (e) and Fig. 2 (a) which gives the equation

$$\Sigma^R_{i,j;\alpha,\beta}(\omega) = -\chi_{i,j;\alpha,\beta}(\omega) = \frac{\lambda^2 g_i \bar{g}_j n_0}{\Delta_a^2} \bar{c}_\alpha c_\beta \sum_{n'} \bigg( \langle \psi_{n'} | \eta_i \bar{\eta}_p | \psi_0 \rangle \langle \psi_0 | \eta_p \bar{\eta}_j | \psi_{n'} \rangle G^R_{\psi,0,n'}(\omega)$$

$$+ \langle \psi_0 | \bar{\eta}_p \eta_i | \psi_{n'} \rangle \langle \psi_{n'} | \bar{\eta}_j \eta_p | \psi_0 \rangle G^A_{\psi,0,n'}(-\omega) \bigg). \quad (23)$$

The term with $G^R$ is described by the diagram Fig. 2 (a) while the second term is from Fig. 9 (e). The density response is then everything inside the summation. Using a constant laser structure factor, real spatial mode functions and inserting the Green's functions from Eq. (22) one arrives at the density response in Eq. (8).

## B  Mode overlaps

To calculate the density response, Eq. (8), one has to compute the overlaps between atom cloud, cavity and laser. In general these overlaps will have to be solved numerical but a closed form solution can be found when the atomic states are well described as eigenstates of the radially symmetric, harmonic trap. Furthermore, the laser form-factor $\eta_p$ also has to be approximated as a constant. With a shallow longitudinal trap and if the atoms are in a zero-temperature BEC then the longitudinal part of the atom eigenstate is tightly localized in momentum space. This means that the longitudinal part of the overlap leads to the atoms scattering into a state with momentum $Q$, set by the cavity geometry. The transverse part of the overlap can be computed both for centred and and non-centred atom clouds. For the more general case when radial symmetry is broken, one finds that the integral over three Hermite polynomials leads to three finite sums with a shared combinatoric pre-factor. However, in the radially symmetric case the result simplifies significantly with the integral to be solved given by

$$\langle \psi_0 | \eta_{j\alpha} | \psi_{n\beta} \rangle = \frac{1}{\pi L_H^2} \sqrt{\frac{j! n!}{(j + |\alpha|)!(n + |\beta|)!}}$$

$$\times \int_0^{2\pi} d\theta \int_0^\infty dr\, r \exp\left[ i\theta(\alpha - \beta) - r^2 \left( L_H^{-2} + \tfrac{1}{2} w_0^{-2} \right) \right] \quad (24)$$

$$\times \left( \frac{r}{L_H} \right)^{|\beta|} \left( \frac{r}{w_0} \right)^{|\alpha|} L_j^{|\alpha|} \left( \frac{r^2}{w_0^2} \right) L_n^{|\beta|} \left( \frac{r^2}{L_H^2} \right),$$

where the mode indices have been split into a radial mode index (Roman letter) and a angular index (Greek letter). The non-BEC scattering state in Eq. (8) leads to a decoupling of cavity modes with different angular index, through the $\delta_{\alpha,\beta}$ from the angular integral. Starting in angular momentum zero we therefore consider only the states with zero angular momentum. The remaining radial integral then has a closed form solution given by

$$\langle \psi_0 | \eta_{j0} | \psi_{n0} \rangle = \langle \psi_{n0} | \eta_j | \psi_0 \rangle = \frac{\Gamma(j + n + 1)}{2^n\,!\,j!} \frac{\left( \delta^2 - \tfrac{1}{2} \right)^j}{\left( \delta^2 + \tfrac{1}{2} \right)^{j+n+1}} {}_2F_1\left[ -n, -j; -n-j; -\frac{\delta^2 + \tfrac{1}{2}}{\delta^2 - \tfrac{1}{2}} \right], \quad (25)$$

where $\delta = w_0/L_H$ is the relative size of the cavity waist compared to the transverse harmonic trapping length, $_2F_1$ is the Gauss hypergeometric function and $\Gamma$ the Gamma function. Clearly these overlaps are fully determined by the cavity waist and radially symmetric harmonic trap strength which sets $L_H$. The simplest case is when the BEC is significantly narrower than the cavity waist. In this limit $\delta^2 \gg 1$ and therefore $\lim_{\delta \to \infty} \langle \psi_0 | \eta_{j0} | \psi_{n0} \rangle = \delta_{n0}$. In all calculations presented in the main text the result from Eq. (25) has been used and the number of atomic states have been truncated only after convergence has been achieved.

## C  Estimating coupling strength between modes

To estimate the effective coupling strength we model the system as having only two modes. We assume the system is far enough away from the critical point such that the Nambu structure is not essential. Furthermore we assume that the crossing happens far enough away from the atom pole such that we can approximate the density response to be constant and real: $\Pi^R(\omega) = \Pi^R$. As we are only looking for an estimate of the coupling strengths we also approximate the two cavity modes to have identical loss rates. Using the self-energy notation from appendix A the retarded Green's function for this system can be written as

$$\mathcal{D}^R(\omega) = \begin{pmatrix} \omega - \Delta_1 + i\kappa - \Lambda^2 \Sigma_{11} & -\Lambda^2 \Sigma_{12} \\ -\Lambda^2 \Sigma_{21} & \omega - \Delta_2 + i\kappa - \Lambda^2 \Sigma_{22} \end{pmatrix}^{-1}, \tag{26}$$

where the different self-energies are real numbers depending on all the microscopic parameters of the full system and $\Delta_i/\kappa$ model the two effective modes of the system. Exactly at an avoided crossing the energies of the two modes coalesce. This can happen if increasing the laser power ($\Lambda$) can bring the two detunings closer to each other, as described in the main text. Exactly at the avoided crossing the laser power satisfies the equation

$$\Delta_1 + \Lambda_{ac}^2 \Sigma_{11} = \Delta_2 + \Lambda_{ac}^2 \Sigma_{22}, \tag{27}$$

where we have defined $\Lambda_{ac}$ to be the specific value of $\Lambda$ that solve the above equation. At this point we define the new parameters

$$\Omega^2 = \Lambda_{ac}^2 \Sigma_{12} = \Lambda_{ac}^2 \Sigma_{21} \text{ and } \tilde{\Delta} = \Delta_i + \Lambda_{ac}^2 \Sigma_{ii}. \tag{28}$$

Using these parameters in $\mathcal{D}^R$ one can write the "11" component of the spectral function (Eq. 12) as

$$A_{11}(\omega) = \frac{\left(\left((\omega - \tilde{\Delta})^2 + \kappa^2\right)\kappa + \Omega^2 \kappa\right)\left((\omega - \tilde{\Delta})^2 + \kappa^2\right)}{\left(\left((\omega - \tilde{\Delta})^2 + \kappa^2 - \Omega^2\right)(\omega - \tilde{\Delta})\right)^2 + \left(\left((\omega - \tilde{\Delta})^2 + \kappa^2\right)\kappa + \Omega^2 \kappa\right)^2}. \tag{29}$$

The peaks in the spectral function emerges when

$$\left((\omega - \tilde{\Delta})^2 + \kappa^2 - \Omega^2\right)(\omega - \tilde{\Delta}) = 0. \tag{30}$$

This has three solutions: $\omega_j \in \left\{\tilde{\Delta}, \tilde{\Delta} \pm \sqrt{\Omega^2 - \kappa^2}\right\}$. At these specific values of $\omega$ the peak height is given by

$$\frac{(\omega_j - \tilde{\Delta})^2 + \kappa^2}{\left((\omega_j - \tilde{\Delta})^2 + \kappa^2\right)\kappa + \Omega^2 \kappa}. \tag{31}$$

We are focusing on the situation where the linewidth is much smaller than the coupling, such that the peaks are well resolved. In this limit the peak height for the $\omega = \tilde{\Delta}$ solution is

proportional to $\kappa/\Omega^2$ while the two other peaks, at $\omega = \tilde{\Delta} \pm \sqrt{\Omega^2 - \kappa^2}$, have identical heights of the order $1/2\kappa$. In the considered regime the solution with the two peaks, separated by twice the coupling strength, clearly dominate the spectral function.

The approach is then to identify an avoided crossing in the system by sweeping the laser power. Once an avoided crossing has been identified the specific laser power, that has two equal-height peaks, is used to define $\Lambda_{ac}$. The coupling strength can then be read off from the splitting between the peaks. Due to all the approximations this is clearly a lowest-order estimate that can easily be improved if a more accurate number is important. For the considered system the approximations are fairly good and even the lowest order estimate will give a result of the right order of magnitude.

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
