# Peer review of "Multimode-polariton superradiance via Floquet engineering"

_SciPost Physics, doi:SciPost Phys. 12, 094 (2022)_

## Round 2 · Referee Report · Anonymous (Referee 1) · 2021-10-28

Strengths

1-This manuscript is well written and organized.
2-This paper goes into great detail on how to implement multimode Floquet polaritons.
3-Overall a good resource for the community actively pursuing experiments with ultracold atomic clouds coupled to optical cavities.

Weaknesses

1- It is unclear what is the selling point of this work. Does the platform described in this paper offer technical advantages over the Floquet polaritons demonstrated in Ref[21]? Does it open an avenue to explore unique physical models that would otherwise be difficult with other multimode platforms? 2- The authors should provide numbers for the achievable coupling rates between the BEC cloud and the optical modes, and effective interactions between the modes, and therefore specify if this system is situated in the mean-field or many-body regime.

Report

In their manuscript, “Multimode-polariton superradiance via Floquet engineering”, Johansen et al. describe an experimental proposal of a multimode optical platform with Floquet engineered interactions for studying many-body states of light. The authors present a system where a cloud of ultracold bosonic atoms is dispersively coupled to many transverse modes of a Fabry-Perot cavity by driving the atomic ensemble using an off-resonant laser with a periodically modulated phase. This work goes into great detail on the effects of phase modulation and careful preparation of these Floquet polaritons, thereby presenting a good resource for the AMO community pursuing BECs coupled to optical cavities.
This area of many-body cavity QED is a topic of great interest. Recent experiments have already demonstrated Floquet polaritons in a twisted cavity [21], a method later used for preparing a Laughlin state of photons [Clark et al. Nature 582, 41–45 (2020)], while others have already observed multimode superradiance in a degenerate confocal cavity [28-31]. It is unclear from reading this paper how this proposal offers any technical advantages over Ref[21] or if it opens a new pathway for exploring new physical phenomena.
While I do have comments along those lines, once these are resolved the manuscript should be published in this journal.

Requested changes

1 - As the authors point out, this work adopts a similar idea to ref. [21] for creating Floquet polaritons, and features a multimode superradiant phase which has been already measured in other systems [28-31]. What is the novelty of the platform being put forward? Does it present a technical advantage over the Floquet system in ref. [21] or does it open a new pathway in exploring different physical models? Can this platform generate stronger mode-mode interactions or offer more flexibility in the number of modes you can couple?

2 - In Fig.5 and 6, the authors show that the modes are coupled from the observed avoided crossings. It would be informative to quantify how strong these atom-mediated multimode interactions are with respect to the polariton lifetimes. This would make the overall story clearer, validating if the system is in the mean-field or many-body regime.

3 - The authors justify the multimode superradiant phase from the renormalization of the critical atom-photon coupling and the appearance of instabilities at finite frequencies. Have the authors explored synchronization effects where the collective coupling leads to coherent emission from an ensemble of transverse modes?

4 - Have the authors considered studying this ordered phase by probing photon correlations g2 between different polariton modes?

5 - Can the authors explain what they mean by ‘deep multimode regime’ (last paragraph of section I), and how the numbers they get for the coupling strengths and interactions apply to this criteria? Are they referring to the fact that the atom-photon coupling is larger than the damping rate for multiple modes? or is the coupling larger than the free spectral range?

6 - The red crosses in Fig.4 are hardly visible due to the colormap, the authors should consider changing the cross colors

  • validity: good
  • significance: good
  • originality: good
  • clarity: ok
  • formatting: good
  • grammar: excellent

Author:  Christian H. Johansen  on 2021-11-15  [id 1944]

(in reply to Report 1 on 2021-10-28)

1 - As the authors point out, this work adopts a similar idea to ref. [21] for creating Floquet polaritons, and features a multimode superradiant phase which has been already measured in other systems [28-31]. What is the novelty of the platform being put forward? Does it present a technical advantage over the Floquet system in ref. [21] or does it open a new pathway in exploring different physical models? Can this platform generate stronger mode-mode interactions or offer more flexibility in the number of modes you can couple?

The two platforms differ fundamentally in the mechanism underlying polariton formation and their interactions. In ref.[21], cavity photons couple to an internal electronic excitation, while in our platform they dispersively couple to the motion of the center of mass of the atom. The polarizability (or susceptibility) of the medium, which defines the properties of the polaritons, is thus very different in the two cases. In our manuscript, we explicitly discuss the linear susceptibility $\chi$ in eq.(7), which is proportional to the density response function of the ultracold atomic cloud. The latter has in general a frequency and momentum dependence which is different from the one shown by the polarizability of an internal electronic excitation. This difference is not restricted to the linear susceptibility, but also to the nonlinear ones. In particular, in the case of ref.[21], the nonlinear susceptibility is dominated by the strong repulsive Rydberg interactions between excited atoms, which have no counterpart in our case. As a net effect, we can for instance expect the stationary and non-stationary superradiant regimes we discuss in section IV to behave very differently, if present at all, in the case of ref.[21].

Furthermore, the discussed driving scheme does give a lot of tuneability. For a specific linear spacing of modes, the coupling of each mode to the atom cloud is determined by the amplitude of the sidebands, which for the single EOM case is determined by the modulation depth. Using a single EOM comes with a trade-off between frequency and modulation depth. Thus, If the modes are not too far apart, one can use a large modulation depth and thereby couple many modes. However, as mentioned in the paper, it is experimentally more efficient to use a multi-frequency drive like in reference [39]. Here they generate a flat frequency comb with a spacing of 250MHz ,which spans the interval between -9GHz up to 9GHz thereby creating 72 sidebands. If a carrier frequency is chosen such that there are supported modes both above and below it, one can make all 73 sidebands couple to a cavity mode that is close to each specific sideband. The result is an equal superposition of 73 modes. The spacing of 250MHz is comparable to the spacing between modes in the Hamburg experiment [23], which is 600MHz. These numbers are meant to give a flavour of the tuneability of the propsal, as the ideal numbers would depend a lot on the specific nature of the cavity modes.

We have added a more thorough comparison with ref. [21] to the introduction.

2 - In Fig.5 and 6, the authors show that the modes are coupled from the observed avoided crossings. It would be informative to quantify how strong these atom-mediated multimode interactions are with respect to the polariton lifetimes. This would make the overall story clearer, validating if the system is in the mean-field or many-body regime.

The total coupling strengths between the modes is given by the pump strength. How this is distributed between the modes is then determined by the modulation depth. Furthermore, each mode can have different detunings which changes the effective coupling between the modes. As a net effect, one can dramatically change the mode coupling by adjusting any one of these parameters.

To take a concrete example consider the few mode case (Fig. 4). To extract effective coupling between two modes one finds the distance between the two peaks in the spectral function, at an avoided crossing, and divides it by two.
For the first avoided crossing at $(\Lambda/\Lambda_c)^2=0.24$, in Fig.4 (b), the effective coupling strength between the $LG_{00}$ and the $LG_{01}$ mode is $g_{LG_{00},LG_{01}}\approx 3.4\kappa$.

Throughout the manuscript we assume that all modes are equally lossy which means that the linear combinations of these also have loss rate $\kappa$ making it a sensible scale. The second avoided crossing involves the avoided crossing between the composite mode, consisting of $LG_{00}$ and $LG_{01}$, and the $LG_{02}$ mode.
As the energy of the composite mode without coupling to the $LG_{02}$ is unknown, the exact position of the avoided crossing is also unknown. This problem can be overcome by investigating the spectral function for the $LG_{02}$ mode ($A_{22}$). In this spectral function only two peaks appear (one for the composite mode and the one for $LG_{02}$) and the effective coupling is proportional to the distance between the two peaks when they have equal height.
This avoided crossing is found at $(\Lambda/\Lambda_c)^2=0.55$ and the effective coupling strength is $g_{LG_{00,01},LG_{02}}=1.27\kappa$. The magnitude of the effective couplings has a strong dependence on the sidebands weight but the exact dependence is non-trivial to extract, as it has to be disentangled from the effects arising from the continuously changing composite nature of the modes. These effects are essential to include as can be seen from the overlaps at p3 in Fig.5

The effective couplings is a quadratic interaction between two different cavity fields and is therefore exactly treatable on a mean-field level. Higher-order processes can lead to effective cavity-cavity interactions which are beyond mean-field. The presented calculations are in a regime where the cavity mediated atom-atom interactions are long range. Therefore, beyond mean-field interaction processes are suppressed, as shown in ref. [45]. In the presented calculation the different coupled cavity modes have an almost identical overlap with the atom cloud and the scaling of the interaction processes simplifies further to scale with the inverse atom number. For experimental realization the number of atoms are usually around $10^4$ to $10^6$ which justifies neglecting these interaction processes.

The discussion of the effective coupling strength has been included in the manuscript in the discussion of Fig.4~(b) in section 3.B.

3 - The authors justify the multimode superradiant phase from the renormalization of the critical atom-photon coupling and the appearance of instabilities at finite frequencies. Have the authors explored synchronization effects where the collective coupling leads to coherent emission from an ensemble of transverse modes? 4 - Have the authors considered studying this ordered phase by probing photon correlations g2 between different polariton modes?

The ordered phase is indeed very interesting but studying it with the same formalism requires a significant extension due to the broken symmetry of the phase. We are therefore currently investigating the ordered phase through a saddle-point approximation of the action (leading to a mean-field equation). Our ongoing investigations actually hint towards the system having stable ordered phases where the composite cavity field oscillates at different frequencies. An essential part of our investigations is to include the fluctuations induced by the cavity loss, and here g2 might be an insightful probe. As these investigations have a very different focus compared to the current manuscript, we will dedicate a separate manuscript to them. We added this point more clearly to the outlook.

5 - Can the authors explain what they mean by ‘deep multimode regime’ (last paragraph of section I), and how the numbers they get for the coupling strengths and interactions apply to this criteria? Are they referring to the fact that the atom-photon coupling is larger than the damping rate for multiple modes? or is the coupling larger than the free spectral range?

We see now that the wording can be misleading and thank the referee for bringing it to out attention. The phrasing 'deep multimode regime' is meant to refer to the fact that a number of modes much larger than 2 is important in the system. After the referees comment we have come to the conclusion that in the specific paragraph the use of 'deep multimode regime' does not add additional clarity and therefore changed it to 'multimode regime'.

6 - The red crosses in Fig.4 are hardly visible due to the colormap, the authors should consider changing the cross colors

We have changed the cross colors as suggested.

---

## Round 3 · Referee Report · Anonymous (Referee 2) · 2021-12-17

Strengths

  • Interacting multimode polaritons are highly topical subject
  • The paper is well organized and very descriptive
  • Creative approach how to realize interacting multimode polaritons using Floquet engineering
  • Gives experimental estimates

Weaknesses

- The mathematical formulation should be more stringent

Report

In this paper the authors consider a cloud of ultracold atoms within a near-planar Fabry-Perot resonator where interactions among the photons are mediated by the atomic excitations. The special point of this proposal is an additional implementation of a pump laser field with a phase modulation which is also periodic in time. The driving frequency of the pump laser is far detuned with respect to the atomic transition. The phase modulation frequency is comparable to the energy difference between the transverse electromagnetic modes of the cavity which gives room to transfer energy from the cavity modes to the Floquet modes. The resulting scattering processes lead to a polarization of the medium and in particular to non-diagonal matrix elements between initial and final cavity modes. These are the interacting polaritons. In the revised version the authors describe how their proposal diverges from pervious ones. The general subject of interacting multimode polaritons is very topical. I find the paper well organized and very descriptive. The experimentally relevant quantities and scales are discussed thoroughly. However, in my opinion the mathematical formulation of the calculation does not reach the same standard. For readers who orientate themselves along the equations there are hurdles from typos and undefined symbols or symbols which are only introduced much later in the text. I also strongly recommend to give more details on the derivation of the most important equations or to quote appropriate references. I will list some examples below but I believe the strong focus on descriptive text with much less rigor in the correct theoretical formulation is present throughout the text. Therefore, before recommending this manuscript for publication in Scipost I would ask to improve on that point. In the following, I will make my criticism more precise in form of explicit examples:

1) In Fig. 1) there are misleading notations. What is \omega_p? Is the \lambda correct in the inset? The figure is discussed in the text in chapter II A while the depth B_m is only introduced in chapter II C. This makes the text and figure confusing. 2) Below Eq. (3) the argument of the spatial mode function should be a vector 3) Eq. 4: alpha is not clarified. 4) How is Q defined in the recoil energy? 5) Give more details on the derivation of Eqs. (7) and (8) from Eq. (6) 6) Below Eq. (8) it should read \bar{c}\alpha c\beta 7) Below Eq. (11) there should be a semicolon separating the indices in the Nambu matrix (Minor typo) 8) The indices in Eq. (12) do not seem to be correct (and should be comma separated as done before). 9) Above Eq. (14): are the indices of the delta functions correct? They do not agree with Eq. (11). 10) Fig 4): The labeling LG_00 on the left top of the figures is confusing as other modes appear in the discussion. Distinguish more clearly between cavity modes and atomic wave modes. The spectral function with only two indices has not been defined before. 11) Eq. (14): the same symbol has been used before with a different meaning. 12) Fig. 5 inset: I is not defined. 13) How are the effective coupling strengths defined? This discussion appears handwaving with no precise theoretical background. 14) It should also be stated in the main text that the laser form factor is taken to be constant. 15) Eq. (A2) the index n^\prime does not seem to be correct.

In conclusion, I am missing a stringent mathematical formulation but if the authors make their concrete calculations more accessible to the interested readers by improving in that direction I believe the paper would be quite substantial and I would recommend publication in Scipost.

  • validity: good
  • significance: high
  • originality: high
  • clarity: good
  • formatting: good
  • grammar: excellent

Author:  Christian H. Johansen  on 2022-01-15  [id 2099]

(in reply to Report 1 on 2021-12-17)

We are very grateful for the referees comments and agree that in several places we had not been careful enough in balancing the mathematical rigour with the descriptive text.
We have carefully gone through the equations and and the referee's comments. The typos have been corrected and slacking notation has been tightened. In places where ambiguities arised we clarified the text. Furthermore, we have added appendix A, which provides a rigorous derivation of Eqs. (7) and (8).
We also included appendix C, that illustrates the precise theoretical background for the coupling-strength estimation procedure used. We hope this clarifies that even though we have taken a descriptive approach to the paper the results presented are rigorously derived.

---

## Round 3 · Referee Report · Anonymous (Referee 1) · 2021-12-21

Report

The authors have addressed my concerns and have made the appropriate changes in the manuscript. I would also like to echo the second reviewer's criticism regarding the mathematical formulation. It is important to make the manuscript more accessible for the readers interested in following these calculations. I am recommending publication in Scipost, but I highly encourage the authors to present their theoretical formulation with more rigor before submitting the final version of the manuscript.

---

## Round 3 · List of Changes

• More thorough comparison with ref. [21] added to the introduction.
  • Discussion of the effective cavity-mode coupling of Fig.4 (b) added to section 3.B.
  • Changed cross colours of fig. 4.
  • Clarified outlook.

---

## Round 4 · Referee Report · Anonymous (Referee 2) · 2022-1-21

Report

In their resubmitted manuscript the authors carefully address all my concerns. In particular adding two new Appendices improved the clarity and comprehension of the theoretical background of the proposal. I now recommend publishing the manuscript in SciPost as it is.

---

## Round 4 · List of Changes

-Added appendix with derivation of polarization and density response
-Added appendix with derivation for estimation of effective coupling strength between cavity modes
-Fixed typos and ambiguities

---

## Editorial Decision

published